# Expression, purification and preliminary pharmacological characterization of the *Plasmodium falciparum* membrane-bound pyrophosphatase type 1

Jianing Liu[1], Keni Vidilaseris[1], Niklas G. Johansson[2,3], Orquidea Ribeiro[1], Loïc Dreano[2], Jari Yli-Kauhaluoma[2], Henri Xhaard[2], Adrian Goldman[1]*

1 Research Program in Molecular and Integrative Biosciences, University of Helsinki, Helsinki, Finland, 2 Drug Research Program, Division of Pharmaceutical Chemistry and Technology, Faculty of Pharmacy, University of Helsinki, Helsinki, Finland, 3 Department of Chemistry, Faculty of Science, University of Helsinki, Helsinki, Finland

* adrian.goldman@helsinki.fi

## Abstract

Membrane-bound pyrophosphatases are integral membrane proteins that catalyze the hydrolysis of pyrophosphate into orthophosphate, while simultaneously facilitating the pumping of protons and/or sodium ions. Since mPPases are absent in humans but play a critical role in the life cycle of protist parasite, they represent promising therapeutic targets. We successfully expressed the *Plasmodium falciparum* type 1 mPPase in the baculovirus/insect cell expression system and purified the protein, yielding 0.3 mg per liter cell culture. Various detergents were tested for solubilization, with the protein remaining active under all selected detergents. *n*-dodecyl-β-D-maltoside combined with cholesteryl hemisuccinate provided the highest solubility (88%). Finally, the PfPPase-VP1 was assayed against a set of fourteen antimalarial drugs, along with seven *Thermotoga maritima* mPPase inhibitors and fourteen compounds of unknown activity against mPPases. Only three compounds, all pyrazolo[1,5-*a*]pyrimidine-based TmPPase inhibitors, retained micromolar $IC_{50}$ activity against PfPPase-VP1. The expression and purification of the PfPPase-VP1 will allow to conduct structural studies as well as to develop target-based screens, two steps necessary for the development of inhibitors to combat parasite disease.

## 1. Introduction

Parasitic protists, such as *Plasmodium* spp., *Leishmania* spp., and *Trypanosoma* spp. are responsible for significant global health challenges. In 2022, approximately 249 million malaria cases were reported [1], along with 0.7–1 million cases of leishmaniasis and 6–7 million individuals affected by Chagas disease [2]. The impact of these diseases is exacerbated by the growing ineffectiveness of current treatments or

**Data availability statement:** All relevant data are within the manuscript and its Supporting Information files.

**Funding:** This work was supported by the Biotechnology and Biological Research Council (BBSRC) (BB/T006048/1) awarded to AG (https://www.ukri.org/councils/bbsrc/), grants from the Academy of Finland (Nos. 1322609 and 1357076) to AG, (No. 308105 and 1355187) to KV and (No. 310297) to HX. (https://www.aka.fi/en/) LJ is funded by the China Scholarship Council (CSC) (https://www.chinesescholarshipcouncil.com/) from the Ministry of Education of P.R. China, whereas NJ was funded by the Finnish Pharmaceutical Society, Magnus Ehrnrooth Foundation and the Swedish Cultural Foundation in Finland. (https://www.suomenfarmaseuttinenyhdistys.fi/en; https://magnusehrnroothinsaatio.fi/en/frontpage/; and https://www.kulturfonden.fi/in-english/). Open access funded by Helsinki University Library (https://www.helsinki.fi/en/helsinki-university-library). The funders had no role in study design, data collection and analysis, decision to publish, or preparation of the manuscript.

**Competing interests:** The authors have declared that no competing interests exist.

their severe side effects [3,4], highlighting an urgent need for new therapeutic strategies to combat these public health crises.

Membrane-bound pyrophosphatases (mPPases) are integral membrane proteins that hydrolyze pyrophosphate into two orthophosphate molecules, a process coupled with the pumping of protons and/or sodium ions across membranes [5–8]. In bacteria, archaea, and *Plasmodium,* these enzymes are typically located in the cell membrane [9,10], while in kinetoplastids, plants, and algae, they are primarily found within acidocalcisomes, vacuoles, and/or the Golgi apparatus [11–13]. Across all organisms where mPPases have been identified, they play a crucial role in regulating various lifecycle processes under diverse stress conditions. In trypanosomatids, mPPases are essential for maintaining ionic gradients across the acidocalcisome membrane, enabling acidocalcisome acidification, cellular adaptation, and parasitic virulence [14]. Knockdown or knockout of mPPase disrupted acidocalcisome acidity and reduced polyphosphate levels, impairing the ability of parasites to transition from the insect vector to the mammalian bloodstream [15–17]. These findings underscore the critical role of mPPases in parasite virulence and their potential as attractive targets for drug development, particularly because mPPases are absent in multicellular animals. mPPases are homodimeric enzymes, with each monomer consisting of 16–17 transmembrane helices (TMHs). The structures of mPPases from *Thermotoga maritima* (TmPPase), *Vigna radiata* (VrPPase), and *Pyrobaculum aerophilum* (PaPPase) in various conformations have been solved using X-ray crystallography [18–20]. In *P. falciparum*, two types of mPPases are present: type 1 (PfPPase-VP1), which is potassium dependent, and type 2 (PfPPase-VP2), which is potassium independent [5]. PfPPase-VP2 is minimally expressed and non-essential in the asexual stages [21], whereas PfPPase-VP1 is highly expressed in the plasma membrane and essential in the ring stage development and the transition to the trophozoite stage of the parasite [5]. Given its essential role in the parasite lifecycle, PfPPase-VP1 represents a promising target for validation as a therapeutic target.

Previously, we used the thermostable TmPPase as a model enzyme for designing inhibitors. A 96-well plate *in vitro* screening assay was developed [22,23], which successfully identified several classes of non-phosphorus inhibitors, including isoxazole- and pyrazolo[1,5-*a*]pyrimidine-based compounds, with low micromolar affinity against TmPPase [24,25]. However, translating these inhibitors into applications requires testing on purified parasitic mPPases, rather than relying solely on TmPPase as a model, as pharmacological differences have been previously demonstrated [26]. Consequently, the overexpression and purification of the PfPPase-VP1 are critical for pharmacological studies. Moreover, purified PfPPase is essential for structural studies, which will provide insights into the atomic-level architecture, facilitating further structure-guided drug discovery.

In this study, we successfully expressed PfPPase-VP1 in *Trichoplusia ni* High Five (Hi5) insect cells, purified the recombinant protein, and assessed its enzymatic activity using the colorimetric molybdenum blue reaction method [27]. Furthermore, the TmPPase activity assay was successfully adapted for PfPPase-VP1 and validated using imidodiphosphate (IDP), a nonhydrolyzable $PP_i$ analogue. A total of 35

compounds were tested for potential PfPPase-VP1 inhibition, including fourteen antimalarial drugs, seven previously identified TmPPase inhibitors, and fourteen compounds readily available in the laboratory with unknown activity on TmPPase.

## 2. Materials and methods

### 2.1. Constructs, cell culture and protein expression

Two constructs of PfPPase-VP1 were designed: one construct fused with GFP and His$_8$-tags at the C-terminus for expression monitoring, and the other tagged with His$_8$ alone at the C-terminus. Both constructs were cloned into the pK509.3 vector [28] and transformed into *E. coli* DH10EMBacY, which contains a baculovirus genome expressing yellow fluorescent protein (YFP) from its backbone in parallel with the heterologous protein target [29]. *Spodoptera frugiperda* (Sf9) and *Trichoplusia ni* High Five (Hi5) cells (Thermo Fisher Scientific) were cultured in suspension at 27°C in Xpress medium (Lonza) and maintained at a density below 3 million cells/mL. Recombinant baculovirus bacmid DNA was transfected into insect cells using X-tremeGENE® HP DNA transfection reagent (Roche) following the protocol provided by the manufacturer. The initial virus (V0) was harvested 60 h post-transfection. Subsequently, 25 mL of 0.5 million Sf9 and Hi5 cells were infected with the V0 virus, and the amplified virus (V1) was harvested after the infected cells reached the proliferation arrest stage (the day after proliferation arrest, DPA). V1 was then used to infect 50 mL of 1 million Sf9 and Hi5 cells. Cell pellets of 100 µL were collected from both cell lines at different time points (DPA, +24 h, +48 h, and +72 h after DPA), resuspended in 1×PBS to a final concentration of 0.9 million cells and then subjected to five freeze-thaw cycles, rapidly freezing in liquid nitrogen and thawing in a 37°C heat block. The samples were analyzed by SDS-PAGE to check for protein expression. The fluorescence intensity of the GFP signals from PfPPase-VP1 was measured using a Sapphire FL (Azure Biosystems, Inc.), normalized against the GFP standard and analyzed with ImageJ (FIJI) software [30]. Stocks of baculovirus-infected Hi5 cells (BIIC) were prepared for large-scale expression as previously described [31], using 1 mL of BIIC per liter scale. Large-scale expression was performed 48 h post- infection.

### 2.2. Membrane extraction and detergent screening

The harvested cells were disrupted using an EmulsiFlex C3 (Avestin) in 30 mL of Lysis Buffer containing 20 mM 4-(2-hydroxyethyl)-1-piperazineethanesulfonic acid (HEPES, pH 7.2), 0.15 M sucrose, 1 mM MgCl$_2$, 50 µM ethylene glycol bis(2-aminoethyl ether)-*N,N,N',N'*-tetraacetic acid (EGTA), 1.33 mM dithiothreitol (DTT), 1.67 mM phenylmethylsulfonyl fluoride (PMSF), and one tablet of Pierce protease inhibitors (Thermo Scientific™). The membrane fraction was isolated by ultracentrifugation at 100,000×g for 1 h and then resuspended in resuspension buffer (50 mM 2-(*N*-morpholino)ethanesulfonic acid (MES)-NaOH (pH 6.5), 20% (v/v) glycerol, 50 mM KCl, 5 mM MgCl$_2$, 1.33 mM DTT, 3 mM pepstatin A, and 1.67 mM PMSF). The total protein concentration of the membrane fraction was measured using the Bradford assay [32]. The extracted membrane was diluted to a total protein concentration of 2.5 mg/mL with resuspension buffer supplemented with 0.03 mM disodium pyrophosphate (Na$_2$PP$_i$) and solubilized in seven different detergents [1% n-dodecyl-β-D-maltoside (DDM), 1% glyco-diosgenin (GDN), 1% lauryl maltose neopentyl glycol (LMNG), 1% octyl glucose neopentyl glycol (OGNG), 1% 4-cyclohexyl-1-butyl-β-D-maltoside (CYMAL4), 1% 5-cyclohexyl-1-pentyl-β-D-maltoside (CYMAL5), and 1% dodecyl maltoside (DDM) + 0.2% cholesteryl hemisuccinate (CHS), w/v]. The solubilization was carried out by incubating the mixture overnight with shaking at 300 rpm at 4°C. The sample was then ultracentrifuged for 1 h at 100,000×g to pellet the undissolved fraction. To estimate the solubilization efficiency of different detergents, samples taken before and after ultracentrifugation were analyzed by SDS-PAGE, and the GFP signals were measured using Sapphire FL. After identifying the optimal detergent, the solubilization time was optimized by mixing the membrane fraction with 1% DDM and 0.2% CHS, supplemented with 0.03 mM Na$_2$PP$_i$, and incubating for various time intervals (1 h, 2 h, 3 h, 4 h, 5 h, 6 h and overnight). The subsequent steps followed the same procedure as in the detergent screening.

## 2.3. Protein purification

10 mL of isolated membrane fraction from a 1-liter culture was solubilized overnight at 4°C with shaking at 300 rpm in solubilization buffer (50 mM MES-NaOH, pH 6.5, 20% glycerol, 1% DDM, 0.2% CHS, 1.33 mM DTT, 3 mM pepstatin A, and 1.67 mM PMSF). After ultracentrifugation at 100,000 × g to remove the pellet, the supernatant was supplemented with 0.07 M KCl, 10 mM imidazole, and 2 mL of Co-Talon beads. The mixture was incubated at 4°C with shaking at 300 rpm for 1 h. Unbound impurities were removed by centrifugation at 3000 rpm for 3 min using a benchtop centrifuge 5804R (Eppendorf). The beads were transferred to a 15 mL Falcon tube and then washed with 2 × column volumes (CV) of washing buffer 1 (50 mM MES-NaOH, pH 6.5, 20% (v/v) glycerol, 50 mM KCl, 40 mM imidazole, pH 6.5, 5 mM $MgCl_2$, 1 mM DTT, 1 mM PMSF, 1 M NaCl and 0.05% DDM or 0.006% GDN or 0.0008% LMNG, depending on detergent used), followed by 6 × CVs of washing buffer 2 (washing buffer 1 without NaCl). The protein was eluted using 2 × CV of elution buffer (50 mM MES-NaOH, pH 6.5, 3.5% (v/v) glycerol, 50 mM KCl, 400 mM imidazole, pH 6.5, 5 mM $MgCl_2$, 1 mM DTT, 1 mM PMSF) with the same detergent as in washing buffer 1. The eluate was then concentrated to 300 μL using Amicon centrifuge concentrators with a 100 kDa cut-off (Millipore EMD), and the imidazole removed using Micro Bio-Spin columns (Bio-Rad). Protein concentration was determined using a NanoDrop Spectrophotometer ND-1000 (Thermo Fisher Scientific) with the Protein A280 module, entering the extinction coefficient and molecular weight of the protein as calculated using the ExPASy ProtParam tool [33].

## 2.4. PfPPase-VP1 activity assay

The activity of PfPPase-VP1 was determined using a molybdenum blue colorimetric activity assay in 96-well plate format to measure the amount of phosphate production, as previously described for TmPPase [22,23,34] but with modifications to pH (5.2–10) and temperature conditions (20–70°C) as described below. To determine the optimal pH for PfPPase-VP1 activity, 5 μL of 2.5 mg/mL total membrane protein was added to the reaction mixture (60 mM of various buffers (below) with pHs ranging from 5.2–10.0, 3 mM $MgCl_2$, 100 mM KCl, and 10 mM NaCl, 100 mM NaF, 5 μM Gramicidin D), to a final volume of 40 μL. Buffers were trisodium-citrate (pH 5.2), MES (pH 6.5), Tris-HCl (pH 6.8, 7.5, 8.0, and 8.8) and glycine-NaOH (pH 10.0). A control sample was prepared by heating the membrane faction at 90 °C for 5 min to inactivate the enzyme. The reaction mixture was incubated at 30 °C for 5 min in a 96-well heating block. After incubation, 10 μl of 2 mM sodium pyrophosphate ($PP_i$) was added, mixed, and incubated for 10 min. The reaction was stopped by adding 60 μL of premixed solution A (0.3 g ascorbic acid in 10 mL of 0.5 M cold HCl) and solution B (70 mg of ammonium heptamolybdate tetrahydrate in 1 mL of cold milli-Q water) and incubating on ice for 1 h. Finally, 90 μL sodium arsenite solution (5 g sodium arsenite, 5 g trisodium citrate dihydrate and 5 mL of iced-acetic acid in 250 mL milli-Q water) was added and incubated at RT for 30 min. Absorbance was measured at 860 nm on a MultiSkan Go UV/Vis spectrophotometer (Thermo Scientific). The specific activity of PfPPase-VP1 in the membrane was calculated using our previously published formula [23]. 60 mM Tris-Cl (pH 7.5) buffer was then selected for optimization of the reaction temperature (RT, 30 °C, 40 °C, 50 °C, 60 °C and 70 °C) by repeating the assay as described above. Ultimately, 60 mM Tris-Cl (pH 7.5) and a temperature of 50 °C were chosen for further activity assays. For the activity assay of purified PfPPase-VP1, 0.15 μg of the enzyme in the optimized detergent solution was mixed with 0.8 μL of 20 mg/mL liposomes (soybean lecithin) and then reaction buffer (60 mM Tris-HCl (pH 7.5), 3 mM $MgCl_2$, 100 mM KCl, and 10 mM NaCl) was added to a total volume of 40 μL. The reaction mixture was preheated at 50 °C for 5 min before adding 10 μl of 2 mM $PP_i$ and mixing. The catalytic reaction was run for 45 min at 50 °C, and the phosphate production was determined as described above. IDP, a known mPPase competitive inhibitor [35], was used as a positive control to validate the assay for inhibitor screening.

## 2.5. In-gel activity assay

An in-native-gel activity assay was conducted to investigate any potential difference in active states or oligomerization of mPPases using the same molybdenum blue colorimetric method described above. Purified PfPPase-VP1 and TmPPase

(as a control) were reconstituted into liposome (soybean lectin) and mixed with sample loading buffer (5% glycerol, 0.01% Ponceau S) for 10 mins. 1-D Clear-native (CN) PAGE was performed using mini-PROTEAN Precast Gels (Bio-rad), as previously described [36]. Electrophoresis was performed at 4 °C with a constant voltage of 100 V for 1 h using 4–12% gradient gels. The gel was then transferred to a container with 50 mL of reaction buffer (60 mM Tris-HCl (pH 7.5), 3 mM MgCl$_2$, 100 mM KCl, and 10 mM NaCl) and placed in an incubator at 50 °C for 5 min. Subsequently, 200 µL of a 10 mM PP$_i$ solution was added, gently mixed and incubated for another 45 min. The container was then placed on ice for 10 mins, and the reaction mixture was removed. To visualize the enzymatic activity product, a mixture of 20 mL of solution A and 2 mL of solution B was added to cover the gel, followed by incubation on ice for another 10 min. Then, 30 mL of sodium arsenite solution was added to the container, and the gel was left at RT for 30 min. Afterward, the sodium arsenite solution was removed, and the gel washed once with 20 mL water before imaging.

### 2.6. Molecular weight determination with SEC-MALS

Size exclusion chromatography coupled with multi-angle light Scattering (SEC-MALS) was conducted to assess the particle distribution and oligomeric state of PfPPase-VP1. The experiments utilize an HPLC system (Shimadzu) equipped with a Superdex 200 10/300 GI column. The flow rate was maintained at 0.3 mL/min at RT, with the SEC buffer composed of 20 mM MES at pH 6.5, 50 mM KCl, 0.006% GDN and 5 mM MgCl$_2$. The MALS system is equipped with MiniDAWN TREOS light scattering and Optilab rEX refractive index detectors (Wyatt Technology Corp). Data processing was performed using ASTRA 6.1 software (Wyatt Technology Corp), and the results were re-plotted using GraphPad Prism 10 for visualization.

### 2.7. Screening for PfPPase inhibitors

All commercial and in-house synthesized compounds used had purities >95% as specified by the vendor and determined by LC–MS and HRMS. A set of fourteen known antimalarial drugs were tested: compounds **1** and **8** were purchased from TCI Europe; **2–4** and **12,** from Sigma Aldrich(from which the free base form **11** was liberated); **5–7,** from Acros Organics; **9** and **10,** from Cayman Chemicals; and **13,** from Fluorochem (from which **14** was liberated in-house to the free base form). A set of seven in-house synthesized TmPPase inhibitors (previously described) were tested on the PfPPase-VP1: compounds **15–16** [24] and **17–21** [37]. The activity of compounds **19–21** against PfPPase-VP1 was published in our earlier work [37]. A set of fourteen compounds with unknown activity were tested against both TmPPase and PfPPase-VP1, including ten isoxazole-based compounds **22–31** and four with other scaffolds **32–35**. Compound **22** was purchased from ChemDiv; **23–28,** from Maybridge; compounds **29–31** were synthesized in-house [38]; and **32–35** were purchased from ChemBridge

The 35 compounds used in this study were pre-dissolved in dimethyl sulfoxide (DMSO) at concentrations of 25–50 mM. For the 96-well inhibition assay, each well contained 0.15 µg reconstituted PfPPase-VP1 in 15 µL of reaction buffer (200 mM Tris-HCl (pH 7.5), 8 mM MgCl$_2$, 333 mM KCl, and 67 mM NaCl) mixed with 25 µL of compound solution diluted in Milli-Q water. Three concentrations (1 µM, 5 µM and 50 µM were used for initial screening, and eight concentrations (0.01 µM, 0.1 µM, 1 µM, 5 µM, 10 µM, 50 µM, 100 µM and 200 µM) were used for potential active compounds in further IC$_{50}$ calculations. The subsequent reaction steps followed the method described in section 2.4 of the PfPPase-VP1 activity assay. The half-maximal inhibitory concentration (IC$_{50}$) of the test compounds was assessed using R 3.6.3 [39] and n-parameter logistic regression (nplr) package [40], following the equation below. $x$ is the logarithm of the inhibitor concentration, $Y$ is the activity value, $B$ and $T$ are the bottom and top asymptotes, respectively, $b$ is the Hill slope, and $x_{mid}$ is the x coordinate at the inflection point.

$$Y = B + \frac{T - B}{1 + 10^{b(x_{mid}-x)})^s}$$

IDP was used in parallel as a positive control. Each concentration of the test compounds was tested in triplicate.

## 3. Results

### 3.1. Overexpression of PfPPase-VP1

We utilized the baculovirus/insect cell expression system to express PfPPase-VP1. To facilitate expression monitoring and purification, PfPPase-VP1 was fused with GFP and a His$_8$ tag. Two insect cell lines (Sf9 and Hi5) were tested for their capacity to express PfPPase-VP1. Both insect cell lines successfully expressed the protein, as indicated by YFP fluorescence in the cytoplasm and GFP fluorescence in the cell membrane, as observed under the microscope (**Fig 1A**). Expression levels in both cell lines reached a plateau by Day3, but Hi5 cells exhibited eight-fold higher PfPPase-VP1 expression compared to Sf9 cells (Fig 1B–C). Given its superior expression yield, the Hi5 cell line was selected as the optimal host for subsequent PfPPase-VP1 production, yielding ~0.3 mg per liter.

### 3.2. Detergent screening

Selecting an appropriate detergent is crucial for membrane protein purification and functionality. Our previous work demonstrated that alkyl maltosides, especially DDM, solubilize 81% of the TmPPase in the membrane [41]. However, despite its efficacy for TmPPase, DDM achieved only a 60% yield when solubilizing PfPPase-VP1 (Fig 2A, C). CHS, a water-soluble cholesterol analog, can create a more native and stable environment for numerous proteins [42] and has been reported to enhance the solubilization of G protein-coupled receptors (GPCRs) when mixed with detergents [43]. Adding CHS to DDM significantly enhanced the solubilization of PfPPase-VP1, achieving an 88% yield. Cymal-4 and Cymal-5, a subset of maltoside detergents with a cyclohexyl aliphatic tail, solubilized approximately 70–80% of PfPPase-VP1. GDN, known for its effectiveness in stabilizing particularly challenging membrane proteins, is often valuable for membrane protein studies [44]. However, it solubilized only about 40% of the PfPPase-VP1 (Fig 2A, C). LMNG, which consists of two DDM molecules, has been shown to outperform DDM in extracting intact membrane proteins and enhancing their stability, especially for GPCRs [45,46]. For PfPPase-VP1, however, the solubilization efficiency of LMNG was as poor as that of GDN (Fig 2A, C). Overall, the DDM + CHS mixture was selected for solubilization given its highest solubilization efficiency. We then optimized the solubilization time, determining that 6 h was sufficient to achieve results

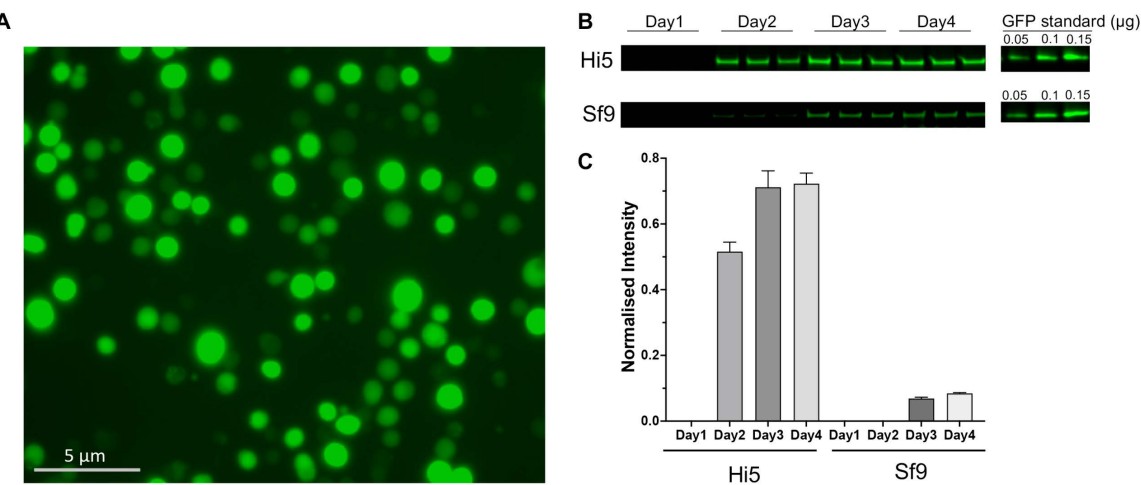

**Fig 1. Baculovirus-transfected Hi5 cells showing increased YFP fluorescence under a laser confocal microscope (20 x magnification).** (A) The fluorescence intensity of YFP at Day3. (B) In-gel fluorescence analysis, imaged using Sapphire FL, illustrating the changes in PfPPase-VP1 expression levels in Hi5 and Sf9 cells at different days post-transfection. Free GFP was used as a control for quantification. (C) The normalized GFP fluorescence intensity of PfPPase expressed in Hi5 and Sf9 cells was calculated using ImageJ.

comparable to overnight solubilization while enabling faster protein purification (Fig 2B, D). Although LMNG and GDN are not effective for solubilizing PfPPase-VP1, they were later used to exchange PfPPase-VP1 from the DDM+CHS detergent mixture during purification to assess their potential for maintaining enzyme activity (see below).

### 3.3. The effect of detergents and constructs on activity

The TmPPase membrane activity assay [41] was optimized for PfPPase-VP1. The maximal activity (~150 µmol/mg/h; Fig 3) was highest at pH 7.5 and 50 °C. Using these conditions, we evaluated the activity of purified PfPPase-VP1 in various detergents (DDM, GDN and LMNG) to determine the detergent that best preserves enzymatic activity (Fig 4). The samples in the three detergents were purified from the same batch, with a dimer-to-monomer ratio of 7:3 on the BN-PAGE (S1 Fig in S1 File). GFP fluorescence and Coomassie stained SDS-page confirmed that the band at an approximate molecular weight (MW) of 100 kDa corresponds to PfPPase-VP1 (Fig 4A). Among the tested detergents, the activity was highest in GDN (142±7 nmol/µg/min), followed by LMNG 120±3 (nmol/µg/min) and DDM/CHS 103±2 (nmol/µg/min) and (p < 0.0001) (Fig 4B).

Our initial construct had a C-terminal GFP tag. Previous studies reported that the C-terminal helix, which is part of the exit channel, plays a role in the hydrolytic cycle and is involved in key interactions monomer-monomer interactions, at least in TmPPase [19]. To determine whether GFP interferes with PfPPase-VP1 activity, we attempted its removal using TEV protease (S2 Fig in S1 File). Despite optimizing the PfPPase-to-TEV ratio, ranging from 30:1–5:1, and testing various incubation conditions (overnight at 4 °C and 2 hours at RT followed by overnight at 4 °C), no significant cleavage of GFP from PfPPase-VP1 was observed. The cleavage region might be inaccessible due to being buried within the detergent micelle. We then expressed and purified GFP-free PfPPase-VP1 with a His$_8$-tag at the C-terminus using the same protocol as described above (Fig 5A). This construct has an approximate MW of 75 kDa (the MW of GFP is around

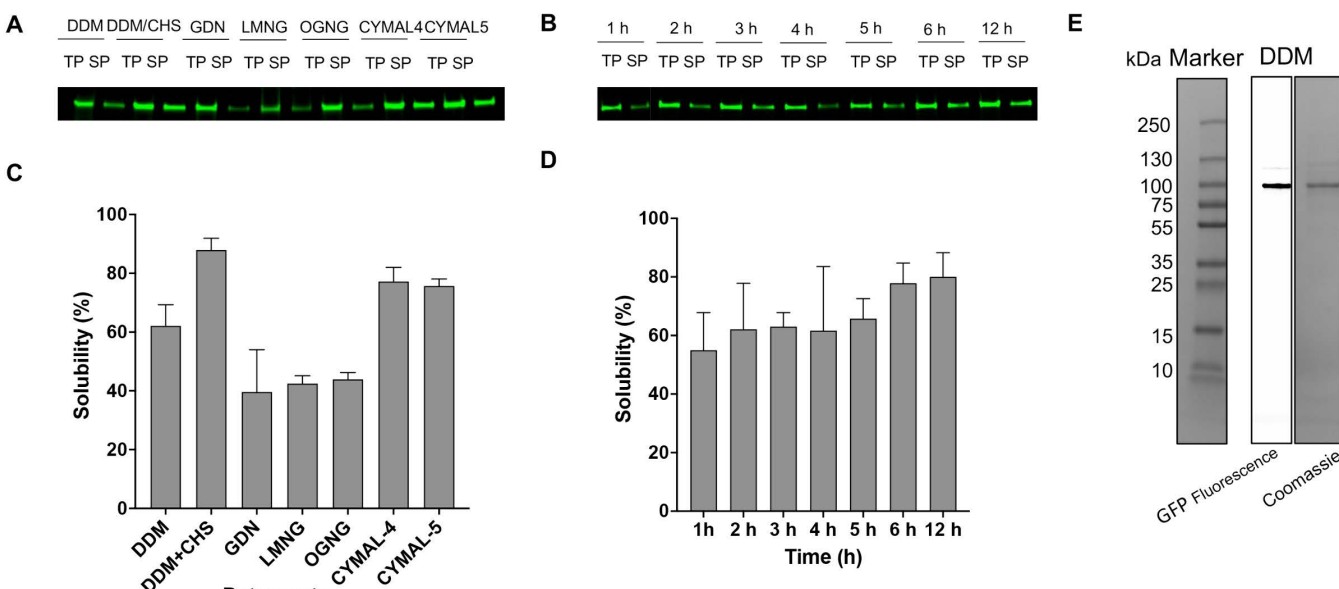

**Fig 2. Detergent solubilization screening of Hi5 expressed His-tagged PfPPase fused with a C-terminal GFP.** Ratio of band intensities of solubilized protein (SP) to total protein (TP) was analyzed using Sapphire FL and solubility percentage was calculated using imageJ. (A) In-gel fluorescence, showing the solubilization efficiency of PfPPase from the following detergents: DDM, DDM+CHS, GDN, LMNG, OG, CYMAL-4, and CYMAL-5. (B) In-gel fluorescence, showing the solubilization efficiency of PfPPase over time for DDM+CHS. (C, D) Relative solubilities of different detergents (C) and as a function of time for DDM+CHS (D).

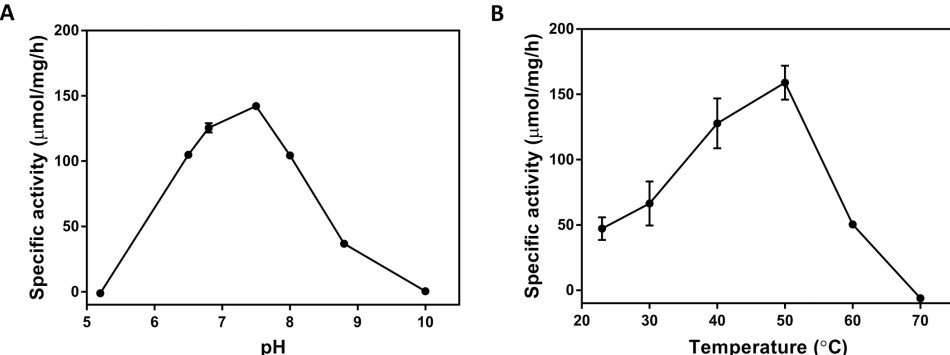

**Fig 3. The effect of pH and temperature on the activity of PfPPase-VP1 membrane.** Specific activity of PfPPase as a function of pH (A) and temperature (B). The optimal pH and temperature were pH 7.5 at 50 °C.

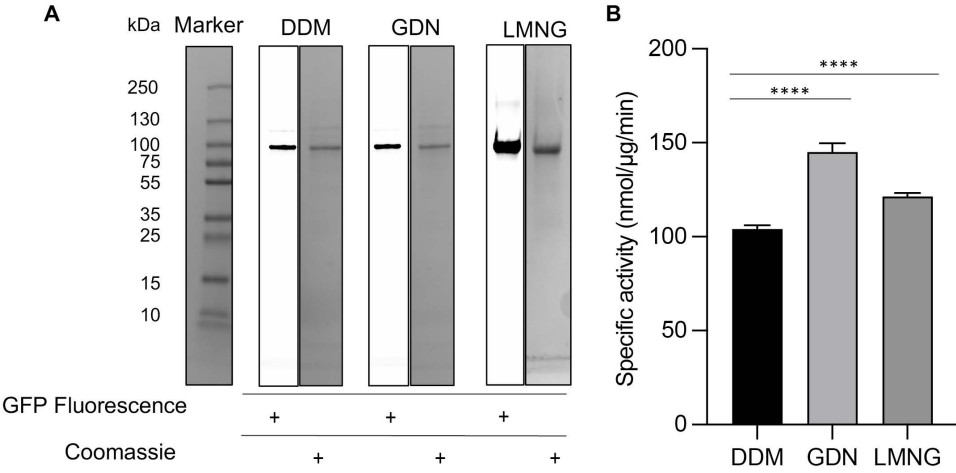

**Fig 4. In-gel fluorescence and Coomassie-stained SDS-page of PfPPase-VP1 in three detergents.** (A) GFP fluorescence and the Coomassie stained SDS-Page showing the purified PfPPase-VP1 in DDM, GDN and LMNG. (B) Bar chart showing the specific activity of PfPPase-VP1 in DDM, GDN and LMNG ($p < 0.0001$). A and B.

27 kDa) and was confirmed as PfPPase-VP1 by anti-mPPase WB. A 96-well plate activity assay revealed that GFP-free PfPPase-VP1 exhibited an activity of $520 \pm 13$ nmol/µg/min (Fig 5B), which is three times higher than that of the GFP-tagged PfPPase-VP1. This finding indicates that the C-terminal GFP tag interferes with $PP_i$ hydrolysis.

### 3.4. In gel activity assay

The primary advantage of the in-gel activity assay is its ability to characterize the activity of enzymes across various oligomeric states. Here, CN PAGE analysis of PfPPase-VP1 revealed the presence of two distinct bands for PfPPase-VP1, whereas TmPPase migrated as a single band (Fig 6A). Molybdenum blue staining confirmed that the activity of PfPPase-VP1 displayed weaker enzymatic activity, with a relative activity of 0.31 compared to TmPPase (Fig 6B-C). Moreover, only the upper PfPPase-VP1 band showed activity, suggesting that it corresponds to the dimeric form, while the lower band represents an inactive monomer PfPPase-VP1. To further investigate the oligomeric state, three independently purified PfPPase-VP1 samples from different expression batches were analyzed by SDS, CN and blue native (BN) pages

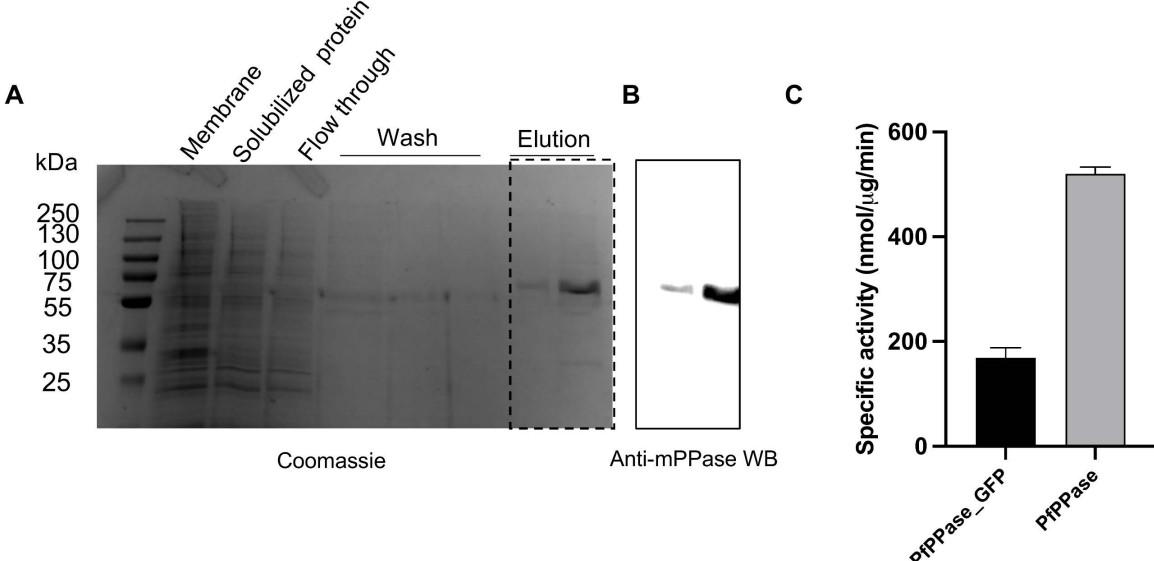

**Fig 5. Purification of PfPPase-VP1 and its activity compared to PfPPase-VP1_GFP.** (A) Coomassie-stained SDS-PAGE image showing the purified PfPPase-VP1 without GFP. (B) Elution fractions from SDS-PAGE (marked with a dotted rectangle), analyzed using anti-mPPase western blot. (C) Bar chart showing the specific activity of PfPPase-VP1 and PfPPase-VP1_GFP, measured at 520 ± 13 nmol/µg/min and 169 ± 19 nmol/µg/min, respectively.

(S3 Fig. **A-C** in S1 File). While a single band was observed on the Coomassie-stained SDS-PAGE, both CN and BN PAGE revealed a distinct secondary lower band between 75 kDa and 130 kDa (S3 Fig. B-**C** in S1 File), indicating the presence of a monomeric form of PfPPase-VP1. The monomer:dimer ratio varied from 8–26% (S3 Fig. **B** in S1 File) Further analysis using SEC-MALS determined the size of PfPPase-VP1/GDN complex to be 450 kDa (S3 Fig. **D** in S1 File). After accounting for the detergent contribution (253 kDa), the calculated MW of PfPPase-VP1 was 176 kDa—significantly lower than the expected 206 kDa for a dimer, confirming the presence of a monomeric fraction. The MW at the leading edge of the PfPPase-VP1 MW curve is 215 kDa, indicating the predominance of the PfPPase-VP1 dimer, while at the trailing edge, the minimum MW of 130 kDa suggests an overlap between monomeric PfPPase-VP1 and its dimer, indicating that they were not completely separated by SEC.

### 3.5. Inhibitor screening

The inhibition assay was designed to evaluate the inhibitory activity of small compounds targeting PfPPase-VP1. IDP served as a positive control, showing an $IC_{50}$ of 175 [137–223] µM against PfPPase-VP1 (Fig 7A). A panel of fourteen anti-malaria drugs was screened for activity against PfPPase-VP1 (Fig 8). Compound 2 showed an $IC_{50}$ value of 190 [156–231] µM (Fig 7B), while the rest of the anti-malarial compounds inhibited neither PfPPase-VP1 nor TmPPase. Previously published compounds 15–21 include three isoxazole-based compounds with $IC_{50}$s below 10 µM and three pyrazolo[1,5-*a*]pyrimidine-based compounds with $IC_{50}$s below 20 µM against TmPPase [26,37]. However, none of the isoxazole-based compounds 15–18 were active against PfPPase-VP1 (Fig 9), while the pyrazolo[1,5-*a*]pyrimidine-based compounds **19–21** showed $IC_{50}$ values of 130, 74 and 58 µM [37] (Fig 7C–E), respectively, which are 4- to 10- fold weaker than against TmPPase [37]. We also screened ten isoxazole-based compounds [**22–31**] and four compounds with other skeletons [**32–35**] that had not previously been tested against TmPPase (S1 Table in S1 File). None of these compounds inhibited PfPPase-VP1. Overall, only compounds **19–21** from the pyrazole[1,5-*a*]pyrimidine core inhibited PfPPase-VP1.

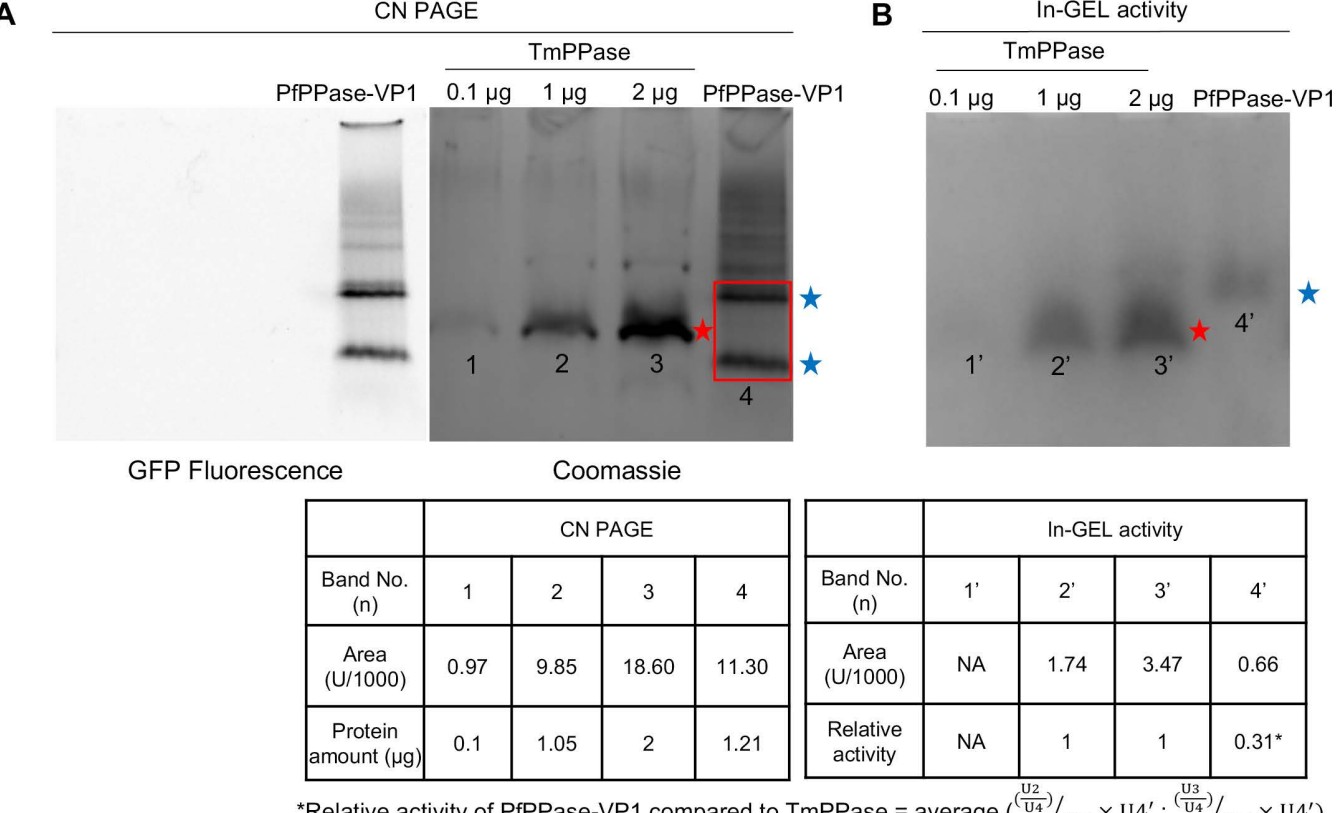

**Fig 6. Clear Native PAGE of PfPPase_GFP and TmPPase.** (A) In-gel fluorescence and Coomassie stained CN PAGE of PfPPase-VP1 and TmPPase showing two oligomeric states of PfPPase-VP1 (blue stars) and the dimeric TmPPase (red star). (B) In-gel activity assay showing the visible complex formed by phosphate with molybdate, corresponding to the position of the upper band of PfPPase-VP1 and TmPPase in (A). (C). Table showing the band intensity measured by ImageJ (Area (U)). Bands correspond to TmPPase [1, 2 and 3] and PfPPase-VP1 [4]. Relative activity of PfPPase-VP1 is normalized against the TmPPase used in the in-gel activity assay (U2 and U3) by ImageJ.

## 4. Discussion

TmPPase has been a useful model system for understanding the structure and mechanism of mPPases, as it comes from a hyperthermophilic organism and so is stable and very easily purified from *S. cerevisiae* by the 'hot-solve' method [47]. We were able to obtain crystal structures of it bound to nonhydrolyzable $PP_i$ analogs, such as IDP, etidronate and zoledronate [18–20] as well as the noncompetitive inhibitor ATC [26]. However, to develop effective inhibitors against parasite mPPases requires expressing parasite mPPases, such as PfPPase-VP1.

As described in this work, we could produce PfPPase in insect cells, but not more related expression systems such as the *Leishmania tarentolae* expression system (data not shown). This is not the first such observation. It has been reported that the enhanced expression of *P. falciparum* reticulocyte-binding protein homolog 5 (PfRh5) and its interacting partner PfRipr in insect cells holds potential for vaccine development as leading candidate antigens [48,49]. We found that Hi5 cells demonstrate superior expression levels compared to Sf9 cells (**Fig 1**). This enhanced expression is likely due to the 3-fold lower release of proteases from Hi5 cells compared to Sf9 cells, leading to reduced protein degradation [50]. Additionally, the larger size of Hi5 cells compared to Sf9 cells may contribute to their higher yields in expressing cell membrane proteins [51].

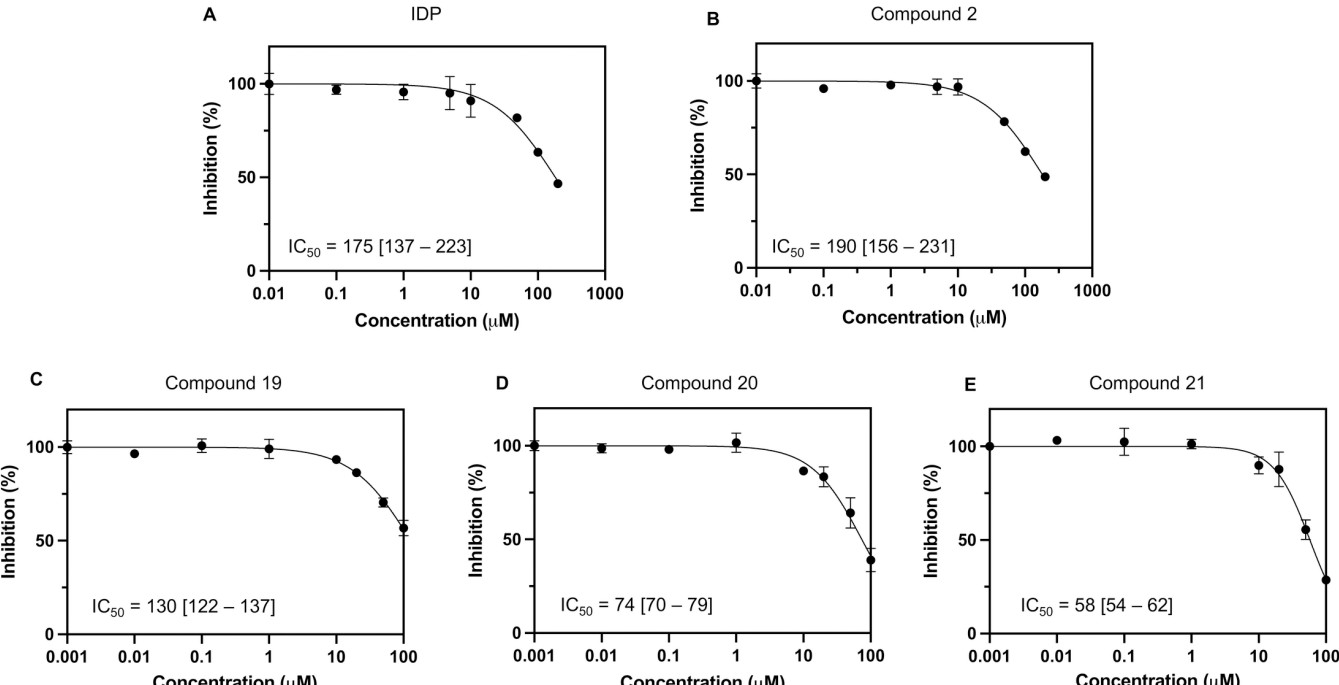

**Fig 7. Concentration-response (IC$_{50}$) for the best PfPPase inhibitors.** (A-E). Inhibition of PfPPase-VP1 by (A) IDP, (B) compound **2**, (D-E) pyrazolo[1,5-*a*]pyrimidine-based compounds **19-21** (D-E). The data for **19-21** has been presented elsewhere [37].

TmPPase exists exclusively as a dimer even after solubilization in DDM, which is consistent with proteins from hyper-thermophilic organisms forming higher oligomeric structures as one means of protein stabilization [52,53]. In TmPPase, the C-terminal main-chain carboxylate of TmPPase forms a salt-bridge with Arg543, meaning that TmPPase is a very stable dimer (S4 Fig in S1 File). PfPPase-VP1, on the other hand, is a heterogeneous mixture, consisting of 8–26% monomeric PfPPase-VP1, alongside its dimeric form (**Fig 6A** and S3 Fig in S1 File). SEC-MALS **(panel D** S3 Fig in S1 File) suggests that the GDN-solubilised PfPPase-VP1 dissociates as it dilutes on the column. The in-gel activity assay revealed that PfPPase-VP1 is active only in its dimeric state (**Fig 6B**), further supporting a half-of-the-sites activity mechanism [54], where inter-subunit communication between two monomers is essential during the catalytic cycle.

PfPPase-VP1 activity is also affected by detergent: it is lower in DDM than in LMNG and GDN (**Fig 4B**). It has been reported that GDN enhances the stability of a few transporters [55], likely due to its stronger detergent-detergent inter-action compared to DDM, which help prevents protein aggregation. LMNG, composed of two DDM molecules, provides higher coverage of the hydrophobic surface of membrane proteins, thereby enhancing their stability [56]. Consequently, replacing DDM with GDN or LMNG proves beneficial for retaining PfPPase-VP1 activity without altering the dimer-to-monomer ratio.

Furthermore, detergents like LMNG and GDN, which have low CMC are desirable in cryo-EM studies, as they help minimize background noise. The CMC of DDM is approximately four and eight times higher than that of GDN and LMNG, respectively [57,58]. The low CMC value of LMNG is advantageous due to its slow off-rate, which allows it to remain bound to the protein surface [46]. In this study, LMNG effectively preserved PfPPase-VP1 activity below its CMC (**Fig 4B**). An extreme case of this phenomenon is observed in the β2 adrenergic receptor, which can bind to its ligand (G-proteins) at concentrations 1000-fold below the CMC of LMNG [59]. Overall, both GDN and LMNG show great potential for maintaining the activity of PfPPase-VP1 and are promising candidates for future cryogenic electron microscopy (cryo-EM) studies.

| Compound | Structure | Drug name | IC$_{50}$ (TmPPase, μM) | IC$_{50}$ (PfPPase, μM) | Vendor /Reference | CAS Number | Purity* (%) |
|---|---|---|---|---|---|---|---|
| 1 | | Artemisinin | Inactive | Inactive | TCI Europe | 63968-64-9 | >97% |
| 2 | | Furamidine dihydrochloride | Inactive | 190 [156–231] | Sigma Aldrich | 55368-40-6 | ≥98% |
| 3 | | Pentamidine isethionate | Inactive | Inactive | Sigma Aldrich | 140-64-7 | ≥98% |
| 4 | | Cycloguanil hydrochloride | Inactive | Inactive | Sigma Aldrich | 152-53-4 | ≥95% |
| 5 | | Mefloquine hydrochloride | Inactive | Inactive | Acros Organics | 51773-92-3 | >97% |
| 6 | | Proguanil hydrochloride | Inactive | Inactive | Acros Organics | 637-32-1 | 97% |
| 7 | | Doxycyline monohydrate | Inactive | Inactive | Acros Organics | 17086-28-1 | 97% |
| 8 | | Atovaquone | Inactive | Inactive | TCI Europe | 95233-18-4 | >98% |
| 9 | | Fosmidomycin | Inactive | Inactive | Cayman Chemicals | 66508-37-0 | ≥95% |
| 10 | | FR900098 (sodium salt) | Inactive | Inactive | Cayman Chemicals | 73226-73-0 | ≥95% |
| 11 | | Chloroquine | Inactive | Inactive | In-house synthesis | 54-05-7 | |
| 12 | | Chloroquine diphospate salt | Inactive | Inactive | Sigma Aldrich | 50-63-5 | ≥98% |
| 13 | | Hydroxychloroquine sulfate | Inactive | Inactive | Fluorochem | 747-36-4 | 95% |
| 14 | | Hydroxychloroquine | Inactive | Inactive | In-house synthesis | 118-42-3 | |

*Commercially obtained compounds had purities ≥95 % as specified by the vendors. Synthesized compounds, as detailed in the references, also showed purities ≥95%.

**Fig 8. Inhibitory activity of anti-malarial drugs against PfPPase and TmPPase.**

| Compound | Structure | Scaffold | IC$_{50}$ (TmPPase, μM) | IC$_{50}$ (PfPPase, μM) | Vendor/Reference | CAS number |
|---|---|---|---|---|---|---|
| 15 | | Isoxazole | 5.4 [5.1-5.7] [26] | Inactive | In-house synthesis [26] | 932508-72-0 |
| 16 | | Other | 33 [29-39] [26] | Inactive | In-house synthesis [26] | 874128-37-7 |
| 17 | | Isoxazole | 6.7 [6.6-6.8] [24] | Inactive | In-house synthesis [24] | 2435664-59-6 |
| 18 | | Isoxazole | 10 [7.3-13] [24] | Inactive | In-house synthesis [24] | 2435664-60-9 |
| 19 | | pyrazolo[1,5-a]pyrimidine | 14 [13-15] [26] | 130 [122-137] [37] | In-house synthesis [26] | 2883115-73-7 |
| 20 | | pyrazolo[1,5-a]pyrimidine | 18 [17-19] [26] | 74 [69.5-78.5] [37] | In-house synthesis [26] | 2883115-77-1 |
| 21 | | pyrazolo[1,5-a]pyrimidine | 14 [13-15] [26] | 58 [54-62] [37] | In-house synthesis [26] | 2883115-75-9 |

**Fig 9. Inhibitory activity of TmPPase inhibitors on PfPPase.** Some of the data, indicated by references, has been presented elsewhere.

GFP fused to the C-terminus of PfPPase-VP1 also resulted in a reduction of its enzymatic activity (**Fig 5B**). Similarly, in TmPPase, fusion of T4L at the C-terminus abolished protein expression [41]. As discussed above, the half-of-the-sites activity mechanism relies on inter-subunit communication, which involves a downward shift of helix 12 and a corkscrew motion of helix 16 [18,19]. Any restriction in the movement of helices 12 and 16 can disrupt the full hydrolysis cycle. For example, the allosteric inhibitor of TmPPase, ATC, binds near the exit channel, interfering with the movement of loop

12–13 and significantly reducing TmPPase activity [26]. In this study, the removal of GFP from the C-terminus resulted in a three-fold increase in activity (Fig 5), likely because the GFP tag, positioned near the exit channel, hinders the motion of surrounding loops or helices. This finding further highlights the significance of conformational rearrangements in the exit channel during the hydrolysis process.

Active PfPPase-VP1 is required for parasite growth [15–17], making it a potential drug target. As we can purify it in mg amounts, testing potential inhibitors *in vitro* is possible. Of the molecules tested, compounds **19** and **20**, derived from the pyrazolo[1,5-*a*]pyrimidine core, show some promise as inhibitor of PfPPase-VP1 [37], with $IC_{50}$ values below 100 µM. However, these compounds exhibit significantly higher potency against TmPPase, with $IC_{50}$ values below 20 µM [25] (Fig 9). Further characterization of the binding sites of compounds **19–21** in TmPPase and PfPPase-VP1 is needed to elucidate the differences in binding affinity between the two enzymes and thus enable the design of better PfPPase inhibitors.

Altogether, this study demonstrates the expression and purification of PfPPase-VP1, as well as our ability to screen inhibitors that target this enzyme. We showed that, surprisingly, the Hi5 insect cell expression system is better-suited for expressing various PfPPase than *Leishmania tarentolae*, even though – unlike that organism – insect cells contain no mPPase orthologues and are not relatively closely related evolutionarily. This work will facilitate future design efforts focused on exploring mechanism or structure of PfPPase-VP1. Future efforts will focus on determining the structure of PfPPase-VP1 through cryo-EM or X-ray crystallography, which will aid in the rational design of more potent and selective inhibitors targeting PfPPase-VP1.

## Supporting information

**S1 File.** **S1 Fig**. BN PAGE of PfPPase-VP1 in three detergents. The BN PAGE showing the ratio of PfPPase-VP1 dimer to monomer in DDM, GDN, and LMNG. The samples were obtained from the same expression batch, and the table presents the percentage of monomeric PfPPase-VP1, measured using ImageJ based on band intensity (Area (U)). **S2 Fig**. Removal of GFP using TEV protease. (A) Coomassie-stained SDS PAGE showing PfPPase-VP1_GFP incubated with TEV under different conditions (overnight at 4 °C; 2 h at RT and then overnight at 4 °C). The band at 25 kDa is TEV protease. (B) In-gel fluorescence of PfPPase-VP1 GFP incubated with TEV under different conditions, showing that GFP remains uncleaved despite different incubation conditions. **S3 Fig**. Expression of PfPPase-VP1 from different batches. (A) GFP fluorescence and Coomassie-stained SDS-PAGE showing purified PfPPase-VP1 from three expression batches. (B). CN PAGE showing the purified PfPPase-VP1 dimer (blue star) and monomer (red star). (C) BN PAGE showing the purified PfPPase-VP1 dimer (blue star) and monomer (red star). The percentage of monomer PfPPase-VP1 were measured using ImageJ based on their band intensity (Area(U)). (D). Characterization of the oligomeric states of PfPPase-VP1 using SEC-MALS. The blue, black and red curves represent the MW of PfPPase-VP1 complex with GDN, the MW of GDN and the MW of PfPPase-VP1, respectively. **S4 Fig**. Salt-bridge interaction between F726 and R543 in TmPPase. (A) Location of the salt-bridge interaction in TmPPase, which stabilizes the TmPPase dimer. (B) Close-up view of two residues (F726 and R543) forming a salt bridge within 3 Å. The distance is shown in yellow dashed line. The TmPPase is shown in cyan, and two residues are shown in purple. **S1 Table 1**. Inhibitory activity of readily available compounds targeting PfPPase and TmPPase
(PDF)

**S1 Raw Images.** **Raw images of all gels and blots presented in the article.**
(PDF)

## Acknowledgments

We thank V. Manole for technical help with SEC-MALS, and N. Sipari (Viikki Metabolomics Unit-Helsinki Institute of Life Science) for mass spectrometry services. The Biocenter Finland DDCB-HiLIFE infrastructure is thanked for organizing resources in the HX laboratory.

## Author contributions

**Conceptualization:** Jianing Liu, Keni Vidilaseris, Henri Xhaard, Adrian Goldman.

**Data curation:** Jianing Liu, Keni Vidilaseris, Niklas G. Johansson, Loïc Dreano.

**Formal analysis:** Jianing Liu.

**Funding acquisition:** Jianing Liu, Keni Vidilaseris, Niklas G. Johansson, Henri Xhaard, Adrian Goldman.

**Investigation:** Jianing Liu, Orquidea Ribeiro.

**Methodology:** Jianing Liu, Keni Vidilaseris, Orquidea Ribeiro.

**Project administration:** Adrian Goldman.

**Resources:** Niklas G. Johansson, Henri Xhaard.

**Supervision:** Keni Vidilaseris, Henri Xhaard, Adrian Goldman.

**Validation:** Jianing Liu, Loïc Dreano.

**Visualization:** Jianing Liu.

**Writing – original draft:** Jianing Liu.

**Writing – review & editing:** Jianing Liu, Keni Vidilaseris, Niklas G. Johansson, Jari Yli-Kauhaluoma, Henri Xhaard, Adrian Goldman.

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
