## [Decision Letter · Decision Letter 0]

15 Apr 2025

PONE-D-25-16760Expression, purification and preliminary pharmacological characterization of the Plasmodium falciparum membrane-bound pyrophosphatase type 1PLOS ONE

Dear Dr. Liu,

Thank you for submitting your manuscript to PLOS ONE. After careful consideration, we feel that it has merit but does not fully meet PLOS ONE’s publication criteria as it currently stands. Therefore, we invite you to submit a revised version of the manuscript that addresses the points raised during the review process. Please submit your revised manuscript by May 30 2025 11:59PM. If you will need more time than this to complete your revisions, please reply to this message or contact the journal office at plosone@plos.org . Please include the following items when submitting your revised manuscript:

We look forward to receiving your revised manuscript.

Kind regards,

Ravinder Kumar, PhD

Academic Editor

PLOS ONE

“This work was supported by the Biotechnology and Biological Research Council (BBSRC) (BB/T006048/1) awarded to AG (https://www.ukri.org/councils/bbsrc/), grants from the Academy of Finland (Nos. 1322609 and 1357076) to AG, (No. 308105 and 1355187) to KV and (No. 310297) to HX. (https://www.aka.fi/en/) LJ is funded by the China Scholarship Council (CSC) (https://www.chinesescholarshipcouncil.com/) from the Ministry of Education of P.R. China, whereas NJ was funded by the Finnish Pharmaceutical Society, Magnus Ehrnrooth Foundation and the Swedish Cultural Foundation in Finland. (https://www.suomenfarmaseuttinenyhdistys.fi/en;
https://magnusehrnroothinsaatio.fi/en/frontpage/; and https://www.kulturfonden.fi/in-english/).”

“This work was supported by the Biotechnology and Biological Research Council (BBSRC) (BB/T006048/1) awarded to AG, grants from the Academy of Finland (Nos. 1322609 and 1357076) to AG, (No. 308105 and 1355187) to KV and (No. 310297) to HX. LJ is funded by the China Scholarship Council (CSC) from the Ministry of Education of P.R. China, whereas NJ was funded by the Finnish Pharmaceutical Society, Magnus Ehrnrooth Foundation and the Swedish Cultural Foundation in Finland. We thank V. Manole for technical help with SEC-MALS, and N. Sipari (Viikki Metabolomics Unit-Helsinki Institute of Life Science) for mass spectrometry services. The Biocenter Finland DDCB-HiLIFE infrastructure is thanked for organizing resources in the HX laboratory.”

“This work was supported by the Biotechnology and Biological Research Council (BBSRC) (BB/T006048/1) awarded to AG (https://www.ukri.org/councils/bbsrc/), grants from the Academy of Finland (Nos. 1322609 and 1357076) to AG, (No. 308105 and 1355187) to KV and (No. 310297) to HX. (https://www.aka.fi/en/) LJ is funded by the China Scholarship Council (CSC) (https://www.chinesescholarshipcouncil.com/) from the Ministry of Education of P.R. China, whereas NJ was funded by the Finnish Pharmaceutical Society, Magnus Ehrnrooth Foundation and the Swedish Cultural Foundation in Finland. (https://www.suomenfarmaseuttinenyhdistys.fi/en;
https://magnusehrnroothinsaatio.fi/en/frontpage/; and https://www.kulturfonden.fi/in-english/).”

Reviewers' comments:

Reviewer's Responses to Questions

**Comments to the Author**

1. Is the manuscript technically sound, and do the data support the conclusions?

Reviewer #1: Yes

Reviewer #2: Yes

2. Has the statistical analysis been performed appropriately and rigorously? 

Reviewer #1: I Don't Know

Reviewer #2: Yes

3. Have the authors made all data underlying the findings in their manuscript fully available?

Reviewer #1: Yes

Reviewer #2: Yes

4. Is the manuscript presented in an intelligible fashion and written in standard English?

Reviewer #1: Yes

Reviewer #2: Yes

5. Review Comments to the Author

Reviewer #1: 1. Yes, Hi5 is clearly giving a higher yield. Data Shows expression analysis was done for four days. Did the authors try a plateau or decline in the expression level. To my understanding it would be good practice to optimize the duration of culture to get an optimal yield.

2. Yes, statistical tools is used but I am not sure if it is used rigorously. (I think the claim should be made carefully for total yield of desired protein and solubility percentage. Particularly in the case of membrane-bound protein). DDM+CHS solubilized more than other used detergents: I hope the authors checked for the maximum extraction of produced protein. I believe some of the protein remains on the membrane and it cannot be resolved during SDS PAGE. Which may affect the image J based determination of protein yield. Hopefully authors have considered it.

3. Yes, the data is available.

4. I believe the manuscript is in presentable form and written in standard English.

Reviewer #2: Manuscript titled" Expression, purification and preliminary pharmacological characterization of the Plasmodium falciparum membrane-bound pyrophosphatase type 1" by Jianing is a nice piece of work. The work is important and done cleanely. Overall the manuscript looks OK but need minor revision before provisionally accepted for publication. My comments and suggestions to authors are as below.

1) Draft need minor writing editing

2) In table 1, 2 please add information about tested compound or drugs like CAS no, grade type, purity % etc

3) In gel image please use white background and bands as black for better visibility

4) In Fig 1 A scale bar is missing

5) Information about protein purity is missing

6. PLOS authors have the option to publish the peer review history of their article (what does this mean? ). If published, this will include your full peer review and any attached files.

**Do you want your identity to be public for this peer review?** For information about this choice, including consent withdrawal, please see our Privacy Policy .

Reviewer #1: No

Reviewer #2: **Yes: ** Dr RAVINDER KUMAR

---

## [Author Response · Author response to Decision Letter 1]

28 Apr 2025

Reviewer #1: 1. Yes, Hi5 is clearly giving a higher yield. Data Shows expression analysis was done for four days. Did the authors try a plateau or decline in the expression level. To my understanding it would be good practice to optimize the duration of culture to get an optimal yield.

Thank you for your comment. As shown in Figure 1, protein expression plateaued on Day 3 (48 hours post-transfection), with no significant increase observed on Day 4 (72 hours post-transfection). We did not extend the experiment to Day 5, as the cells began to break down and viability declined noticeably (see figure below). This could decrease the quality of the expressed protein. Based on these observations, we selected Day 4 as the optimal harvest time.

2. Yes, statistical tools is used but I am not sure if it is used rigorously. (I think the claim should be made carefully for total yield of desired protein and solubility percentage. Particularly in the case of membrane-bound protein). DDM+CHS solubilized more than other used detergents: I hope the authors checked for the maximum extraction of produced protein. I believe some of the protein remains on the membrane and it cannot be resolved during SDS PAGE. Which may affect the image J based determination of protein yield. Hopefully authors have considered it.

We acknowledge the reviewer concern regarding the determination of solubility percentage.Yes, we took this into careful consideration. To isolate the solubilized protein from the membrane as effectively as possible and to maximize protein yield, we performed ultracentrifugation at 100,000 ×g for 1 hour. A similar method was also used by Liu et al., to screen detergent and calculate protein yield (Liu et al., Methods Mol. Biol.2020). This protocol was consistently applied to all detergents screened, ensuring a fair comparison of their solubilization efficiency. Therefore, we believe the reported percentage of solubilized protein is reliable across all detergents tested.

3. Yes, the data is available

4. I believe the manuscript is in presentable form and written in standard English.

Reviewer #2: Manuscript titled" Expression, purification and preliminary pharmacological characterization of the Plasmodium falciparum membrane-bound pyrophosphatase type 1" by Jianing is a nice piece of work. The work is important and done cleanely. Overall the manuscript looks OK but need minor revision before provisionally accepted for publication. My comments and suggestions to authors are as below.

1) Draft need minor writing editing

Thank you for your input. We have revised the draft based on the reviewers suggestions.

2) In table 1, 2 please add information about tested compound or drugs like CAS no, grade type, purity % etc

We have added additional information about the compounds used in Tables 1 and 2 of the manuscript.

3) In gel image please use white background and bands as black for better visibility

We apologize for the confusion. We have reversed the backgroud from black to white in the GFP fluoresence SDS PAGE images.

4) In Fig 1 A scale bar is missing

Thanks for pointing this out, now we added the scale bar on the Fig 1A.

5) Information about protein purity is missing

Thanks for pointing this out. SDS-PAGE analysis showed a single prominent band corresponding to the expected molecular weight of the target protein, with no detectable impurity bands, suggesting a purity level of >95%. We have included the result in the manuscript.

---

## [Editor Report · Decision Letter 1]

2 May 2025

Expression, purification and preliminary pharmacological characterization of the Plasmodium falciparum membrane-bound pyrophosphatase type 1

PONE-D-25-16760R1

Dear Dr. Jianing Liu

We’re pleased to inform you that your manuscript has been judged scientifically suitable for publication and will be formally accepted for publication once it meets all outstanding technical requirements.

Kind regards,

Ravinder Kumar, PhD

Academic Editor

PLOS ONE
---

## [Editor Report · Acceptance letter]

PONE-D-25-16760R1

PLOS ONE

Dear Dr. Liu,

I'm pleased to inform you that your manuscript has been deemed suitable for publication in PLOS ONE. Congratulations! Your manuscript is now being handed over to our production team.

Kind regards,

on behalf of

Dr. Ravinder Kumar

Academic Editor

PLOS ONE